# A stable mode of bookmarking by TBP recruits RNA polymerase II to mitotic chromosomes

Sheila S Teves[1]*, Luye An[2], Aarohi Bhargava-Shah[1], Liangqi Xie[1], Xavier Darzacq[1], Robert Tjian[1,3]*

[1]Department of Molecular and Cell Biology, Li Ka Shing Center for Biomedical and Health Sciences, CIRM Center of Excellence, University of California, Berkeley, Berkeley, United States; [2]Department of Biochemistry, Molecular and Cell Biology, Cornell University, Ithaca, United States; [3]Howard Hughes Medical Institute, Berkeley, United States

**Abstract** Maintenance of transcription programs is challenged during mitosis when chromatin becomes condensed and transcription is silenced. How do the daughter cells re-establish the original transcription program? Here, we report that the TATA-binding protein (TBP), a key component of the core transcriptional machinery, remains bound globally to active promoters in mouse embryonic stem cells during mitosis. Using live-cell single-molecule imaging, we observed that TBP mitotic binding is highly stable, with an average residence time of minutes, in stark contrast to typical TFs with residence times of seconds. To test the functional effect of mitotic TBP binding, we used a drug-inducible degron system and found that TBP promotes the association of RNA Polymerase II with mitotic chromosomes, and facilitates transcriptional reactivation following mitosis. These results suggest that the core transcriptional machinery promotes efficient transcription maintenance globally.

DOI: https://doi.org/10.7554/eLife.35621.001

*For correspondence:
sheila.teves@berkeley.edu (SST);
jmlim@berkeley.edu (RT)

## Introduction

The cell cycle presents a challenge to maintenance of transcription programs. First, the genome becomes highly condensed (*Koshland and Strunnikov, 1996*). Secondly, cell cycle dependent phosphorylation cascades inactivate transcription globally (*Prescott and Bender, 1962*; *Rhind and Russell, 2012*; *Taylor, 1960*). Lastly, the disassembly of the nuclear membrane combined with TF exclusion from mitotic chromosomes was thought to result in the dispersal of most TFs throughout the cytoplasm (*Gottesfeld and Forbes, 1997*; *John and Workman, 1998*; *Martínez-Balbás et al., 1995*; *Rizkallah and Hurt, 2009*). How then do the new daughter cells faithfully re-establish the original transcription program? The observation that DNase I hypersensitive sites were maintained in mitotic chromosomes (*Martínez-Balbás et al., 1995*) led to the theory of 'mitotic bookmarking' (*Michelotti et al., 1997*) where a select group of TFs maintain the ability to bind to target sequences during mitosis and 'bookmark' target genes for efficient reactivation. The first few TFs identified to bind to mitotic chromosomes provided early support for the bookmarking theory (*Caravaca et al., 2013*; *Kadauke et al., 2012*; *Xing et al., 2005*). Recently, we and others discovered that previous evidence for the widely observed exclusion of TFs from mitotic chromosomes was primarily due to formaldehyde crosslinking (*Lerner et al., 2016*; *Pallier et al., 2003*; *Teves et al., 2016*). In fact, most TFs remain associated with mitotic chromosomes but in a dynamic manner. Rather than acting as a stable bookmark, TFs seem to 'hover' on mitotic chromosomes. However, it was not clear how

such a dynamic mode of bookmarking by TFs could maintain stable transcriptional memory following mitosis.

To effect transcriptional regulation, TFs at enhancers must cross-talk with the RNA Polymerase II (Pol II) machinery at promoters. Previous studies have shown that certain transcription start sites (TSSs) remain sensitive to permanganate oxidation during mitosis (*Michelotti et al., 1997*), suggesting the potential for a promoter-specific bookmarking mechanism. Central to the recruitment of Pol II to promoters is the TATA-binding protein (TBP), the prototypic member of the multi-subunit core promoter recognition ensemble TFIID (*Goodrich and Tjian, 1994*), making TBP an attractive candidate for a global facilitator of transcriptional memory. Indeed, a number of studies have tested the hypothesis of TBP as a mitotic bookmarker. For instance, immunofluorescence (IF) staining of TFIID subunits, including TBP, showed cellular redistribution of the subunits away from chromosomes during mitosis but that a subpopulation of TFIID subunits, including TBP, remains associated with biochemically purified mitotic chromosomes (*Fairley et al., 2003*; *Segil et al., 1996*; *Teves et al., 2016*). Furthermore, live-cell imaging of GFP-tagged but over-expressed TBP shows enrichment on chromosomes throughout mitosis (*Chen et al., 2002*). We now know, however, that some of these experiments need to be revisited, as these early studies suffered from potential experimental biases, including unanticipated TF exclusion induced by formaldehyde crosslinking as well as over-expression biases for proteins with highly regulated concentrations within the cell (*Lerner et al., 2016*; *Pallier et al., 2003*; *Teves et al., 2016*; *Um et al., 2001*). Furthermore, some studies have found that TBP remains bound at specific loci during mitosis via chromatin immunoprecipitation (*Christova and Oelgeschläger, 2002*; *Xing et al., 2008*), whereas others have not (*Kelly et al., 2010*), though differences may be cell-type specific. More importantly, the functional consequence of TBP bookmarking (i.e. promoting transcriptional memory through mitosis), if any, has thus far not been established.

Through live-cell imaging of endogenously tagged proteins in mouse embryonic stem cells (mESCs), we report that TBP stably binds to interphase chromatin for minutes. Moreover, TBP maintains a stable association with mitotic chromosomes, and occurs on a global scale at promoters of active genes. Using an endogenous knock-in of a drug inducible protein degradation system, we found that TBP recruits a small fraction of Pol II molecules to mitotic chromosomes, and that a subset of the recruited Pol II retain activity in mitosis. Lastly, nascent RNA analysis indicates that TBP promotes efficient reactivation of global transcription programs following mitosis.

## Results

### Endogenous TBP bookmarks mitotic chromosomes

Previous studies have shown that mitotic chromosomes globally retain accessibility (*Hsiung et al., 2015*; *Teves et al., 2016*). Focusing specifically at the TSS of mESCs, we analyzed our previously published ATAC-seq data for all genes. Reads under 100 bp indicate potential transcription factor (TF) binding sites (*Buenrostro et al., 2013*). Short reads are highly enriched at the TSS of active genes in asynchronous cells and maintained in mitosis (*Figure 1A*, *Figure 1—figure supplement 1*), suggesting that TSSs may be bookmarked by promoter bound factors. We also analyzed the mono-nucleosome sized fragments (180–250 bp) from the ATAC-seq at the TSS genome-wide. Remarkably, we found that the accessibility of the nucleosomes flanking the TSS increased during mitosis relative to interphase chromatin (*Figure 1B*, *Figure 1—figure supplement 1*). These results suggest that nucleosomes around the TSS may assume a different configuration during mitosis than in interphase that allows for increased transposition activity. One possibility is that the global decrease in transcription observed in mitosis may result in decreased occupancy and/or decreased residence time of trans-acting factors at the TSS, resulting in an apparent increase in accessibility of the surrounding nucleosomes.

Given the persistence of accessibility at promoters of mitotic mESCs and previous reports of TBP binding in mitosis (*Chen et al., 2002*; *Segil et al., 1996*; *Teves et al., 2016*), we set out to visualize TBP dynamics throughout the cell cycle in mESCs. We used CRISPR/Cas9 to knock-in HaloTag at the endogenous TBP locus and allow for live-cell imaging of endogenous proteins, which remain functional (*Figure 1—figure supplement 2*). After stable integration of H2B-GFP in the Halo-TBP C41

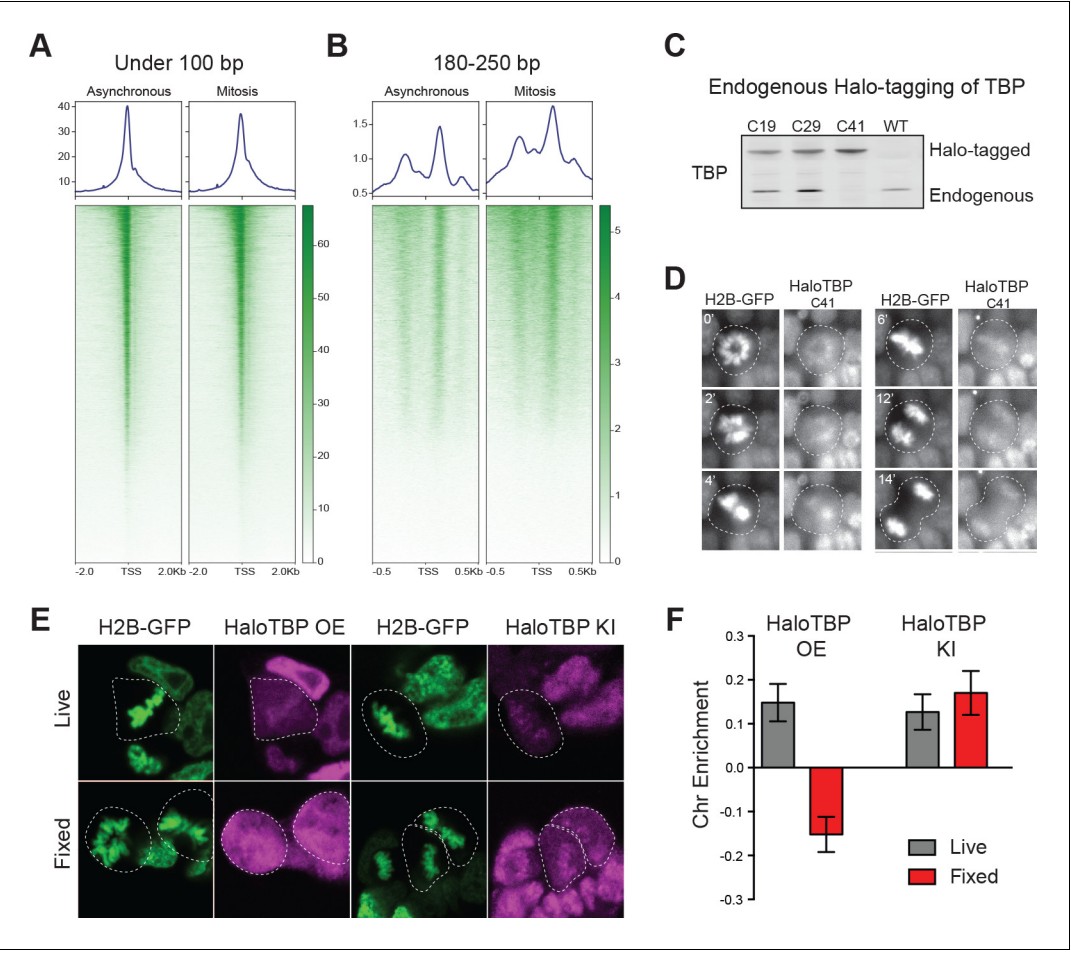

**Figure 1.** Endogenous TBP associates with mitotic chromosomes. (A–B) Previous ATAC-seq data (*Teves et al., 2016*) were examined for TSS analysis. Heatmaps for all genes centered at the TSS (bottom) and aggregate plots of global average signal (top) surrounding the TSSs of all genes were generated for reads under 100 bp (A), and for mononucleosome-sized fragments (B). (C) Western blot analysis of whole cell extracts of clones derived from endogenous tagging of TBP to insert the HaloTag. (D) Time-lapse live-cell imaging of C41 HaloTBP cells stably expressing H2B-GFP as cells undergo mitosis. (E) Live imaging versus fixed immunofluorescence for cells over-expressing Halo-TBP or the endogenous C41 Halo-TBP knock-in. (F) Chromosome enrichment levels for either the over-expressing Halo-TBP cells under live or fixed conditions, or the C41 Halo-TBP knock-in cells under live or fixed conditions. n = 40 cells. Data are represented as mean ± SEM.

DOI: https://doi.org/10.7554/eLife.35621.002

The following figure supplements are available for figure 1:

**Figure supplement 1.** Accessibility at the TSS is maintained in mitosis.
DOI: https://doi.org/10.7554/eLife.35621.003

**Figure supplement 2.** Halo-tagged TBP mouse ES cell line remains pluripotent.
DOI: https://doi.org/10.7554/eLife.35621.004

homozygous line, we observed that the endogenous Halo-TBP is enriched on mitotic chromosomes throughout mitosis (*Figure 1D*).

To test whether over-expressed Halo-TBP could recapitulate the behavior of endogenous Halo-TBP we analyzed the enrichment on mitotic chromosomes of stably over-expressed (Halo-TBP OE) and the endogenous knock-in (Halo-TBP KI) in live versus fixed cells as previously described (*Teves et al., 2016*). Under live imaging conditions, both knock-in and overexpressed TBP show enrichment on mitotic chromosomes (*Figure 1E,F*). However, fixation led to eviction of overexpressed but not the knock-in, suggesting that over-expression may not fully recapitulate the complex endogenous regulation of TBP, such as the ubiquitination/deubiquitination process that tightly

regulates TBP levels in cells (*Li et al., 2015*). Our observation that endogenous TBP is insensitive to fixation-based exclusion from chromosomes, in contrast to most sequence-specific TFs, suggests a distinct interaction modality between TBP and mitotic chromosomes.

## Endogenous TBP stably binds to mitotic chromosomes

To examine TBP interaction with mitotic chromosomes, we imaged individual Halo-TBP KI molecules at a frame rate of 133 Hz and high illumination intensity using stroboscopic photoactivatable single particle tracking (spaSPT) (*Hansen et al., 2017*, *2018*). We then localized individual particles and measured their displacement between consecutive frames (*Figure 2—figure supplement 1*). Short displacements are indicative of bound molecules whereas long displacements are suggestive of diffusing ones. The shift to longer displacements in mitosis compared to interphase (*Figure 2A*) suggests a difference in the bound fraction and/or the diffusion constant between the two cell cycle stages. We then used Spot-On, an analysis tool for kinetic modeling of SPT data, and fitted the distribution of displacements (*Figure 2C*) (*Hansen et al., 2018*). A 2-state kinetic model representing 'bound' and 'free' populations could not adequately fit the data (*Figure 2—figure supplement 2*), suggesting that a dual classification of 'bound' and 'free' TBP populations may be insufficient to accurately characterize the data. Therefore, we fitted a 3-state kinetic model to represent three distinct populations: a fast-diffusing TBP population ('fast'), a slower diffusing population wherein TBP may be diffusing in a complex with the TAFs as TFIID ('slow'), and a stably DNA bound population ('bound') (*Figure 2B*). The 3-state model fitted the data more precisely (*Figure 2A*, *Figure 2—figure supplements 2*). From this analysis, we estimate that in interphase 27.1% of TBP molecules are bound, whereas 13.3% are bound in mitotic cells (*Figure 2C*). This decrease in fraction bound during mitosis may reflect the minimum amount of TBP needed to 'bookmark' active genes during mitosis.

We next examined how stable is TBP binding to chromatin by performing single particle tracking (SPT) with long exposure times (500 ms) as previously described (*Watanabe and Mitchison, 2002*). This imaging technique allows for a 'blurring out' of diffusing molecules while bound ones appear as diffraction-limited spots. After localization and tracking, we measured the dwell time, the amount of time each molecule remains detected, and plotted the log-log histogram of dwell times for interphase and mitotic cells (*Figure 2D*). We then fitted a two-component exponential decay model, representing specific versus non-specific binding events, to the dwell time histograms, and extracted the photobleaching-corrected residence times for the two populations (*Hansen et al., 2017*; *Teves et al., 2016*). The residence time for specific Halo-TBP binding during interphase is on average 88 s (*Figure 2E*). This binding is significantly more stable than typical sequence-specific TFs like Sox2, p53, GR, which normally have residence times on the order of a few to around ten seconds in interphase (*Chen et al., 2014*; *Mazza et al., 2012*; *Mueller et al., 2008*; *Normanno et al., 2015*; *Swinstead et al., 2016*). Halo-TBP also remains stably bound during mitosis (*Figure 2E*), with a calculated average residence time of 118 s. The longer calculated residence time is likely due to the small fraction of TBP molecules (less than 1%) with very long dwell times (above 100 s in *Figure 2D*). Given that the whole distribution profiles appear very similar, we concluded that for most of the bound TBP, residence times are indistinguishable between interphase and mitotic cells. This result is in stark contrast to Sox2, whose residence time decreases by at least half during mitosis (*Teves et al., 2016*), suggesting that TBP binding to mitotic chromosomes may be independent of transcriptional activity.

As an orthogonal approach to SPT, we performed FRAP (Fluorescence Recovery after Photobleaching) analysis to independently measure TBP binding dynamics during interphase and mitosis. We plotted the FRAP recovery at the bleach spot over time for Halo-TBP in interphase and in mitotic cells, along with Halo-3xNLS and H2B-Halo (*Figure 2F*). When normalized for bleach depth, the recovery curves for interphase and mitotic cells become super-imposable (*Figure 2G*). The difference in bleach depth is consistent with a decrease in the fraction of bound molecules between interphase and mitotic cells, whereas the super-imposable recovery curves is consistent with similar residence times of the bound fraction.

## TBP binds to the TSS of active genes in mitosis

Given that TBP is resistant to formaldehyde-based exclusion, we performed crosslinked ChIP-seq analysis to determine where TBP binds in mitosis. Highly reproducible ChIP-seq replicates show that

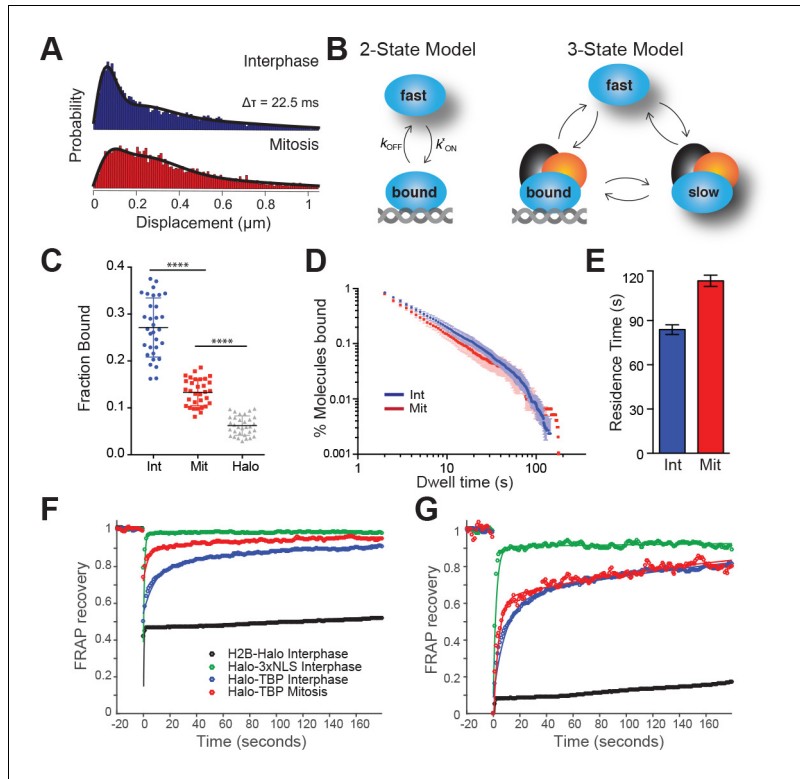

**Figure 2.** TBP dynamics within live cells. (**A**) Cells were labeled with 25 nM pa-JF549 and individual molecules were tracked over 25,000 frames. Jump length histogram measured in displacements (µm) after three consecutive frames ($\Delta\tau$ = 22.5 ms) during spaSPT for cells in interphase (blue) or in mitosis (red). (**B**) Depiction of different states for the 2-state or the 3-state kinetic model. For the 2-state model, the molecules switch from fast diffusing mode to bound states. In the 3-state model, the freely diffusing molecules can be divided into two categories, fast and slow, which can switch to a DNA-bound state. (**C**) Scatter plot of fraction bound from individual cell model fits from interphase (Int) and mitotic cells (Mit). mESCs stably expressing HaloTag only (Halo) were imaged and the fraction bound for each individual cell was also extracted from model fitting to show non-specific binding population. n = 32 cells over four biological replicates. ****p-value<0.0001 (**D**) Dwell time histogram of the fraction of endogenously-tagged Halo-TBP molecules remaining bound for interphase (blue) and mitotic (red) cells. (**E**) Quantification of the residence time of Halo-TBP in interphase (blue) and mitotic (red) cells. n = 30 cells. (**F**) Quantification of fluorescence recovery at the bleach spot for the indicated Halo-tagged construct in interphase and mitosis. n = 30 cells. (**G**) Data from F, normalized for bleach depth. Data are represented as mean ± SEM.
DOI: https://doi.org/10.7554/eLife.35621.005

The following figure supplements are available for figure 2:

**Figure supplement 1.** Halo-TBP shows increased movement in mitosis relative to interphase.
DOI: https://doi.org/10.7554/eLife.35621.006

**Figure supplement 2.** Model fitting of Halo-TBP fast tracking data for interphase and mitotic cells.
DOI: https://doi.org/10.7554/eLife.35621.007

TBP is maintained at largely the same genomic sites in mitosis as in interphase chromatin (*Figure 3—figure supplements 1* and *2*). After peak calling in the combined replicates, we plotted the $\log_2$ normalized reads in binding sites for asynchronous and mitotic samples and found that while there is a slight decrease in the average number of reads in binding sites for mitotic samples, the peak counts are largely very similar (*Figure 3A*). We then performed differential peak analysis using DiffBind (*Ross-Innes et al., 2012*; *Stark and Brown, 2011*), and identified only 374 out of 60,591 peaks that are considered differentially enriched (*Figure 3B*). Out of this, 72 peaks have higher enrichment in mitotic cells while 302 peaks show higher enrichment in asynchronous samples (*Figure 3A*). This analysis shows that the majority of the TBP bound sites are largely maintained in both asynchronous and mitotic cells.

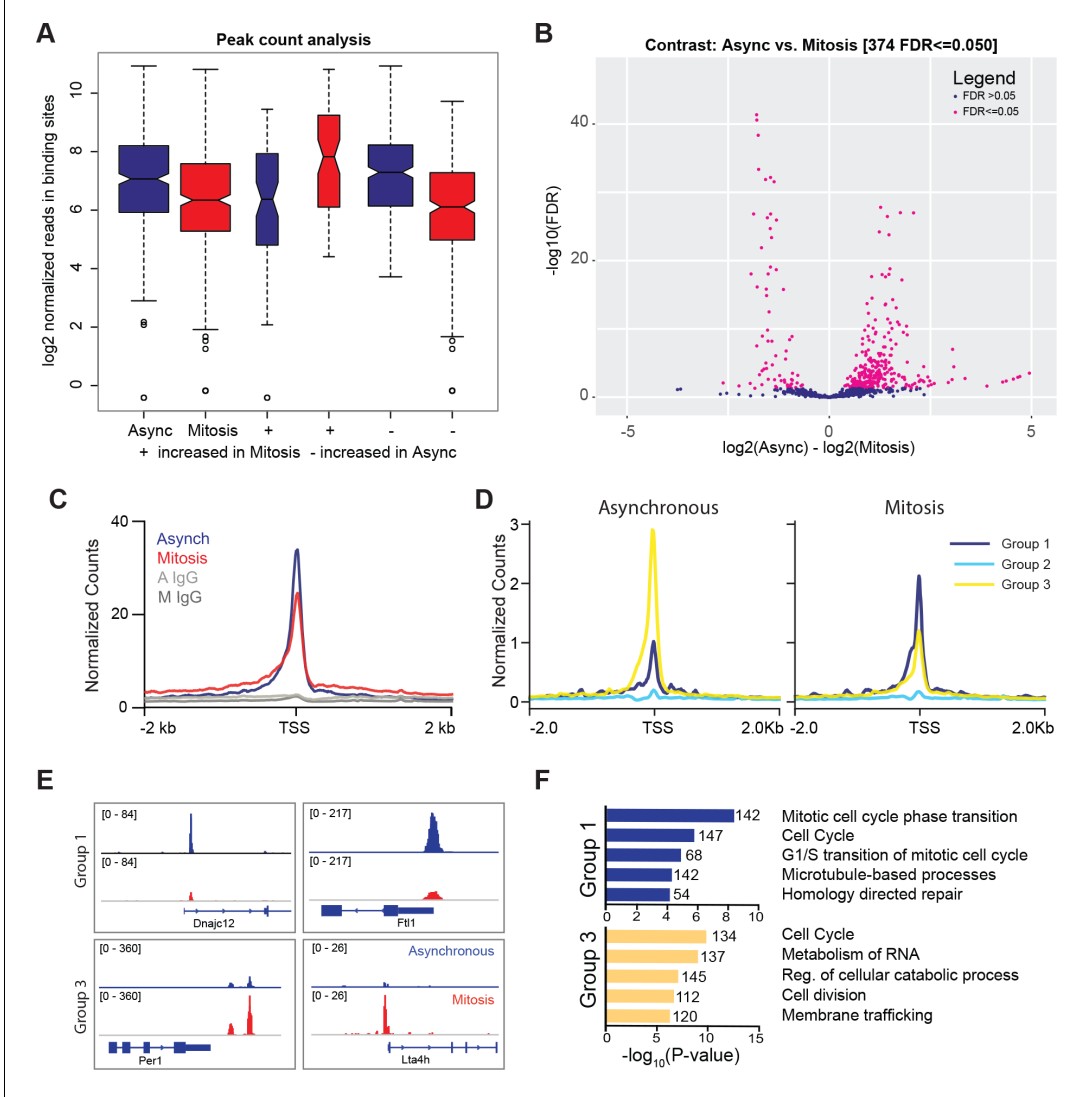

**Figure 3.** TBP maintains binding to promoters of active genes during mitosis. (**A**) Box plot of read distributions for all identified peaks. Asynchronous and mitotic samples are shown in blue and red, respectively. The first two boxes represent all peaks, the middle two are peaks that have higher levels in mitosis, and the last two boxes are peaks that have higher reads in asynchronous samples. (**B**) Volcano plot of all peaks in asynchronous and mitotic samples. Peaks identified as differentially bound are shown in pink (n = 374). (**C**) Average ChIP-seq read density for all TSS and surrounding regions for asynchronous (blue) and mitotic (red) samples and corresponding IgG controls (grays). (**D**) Unbiased k-means clustering of data from B with k = 6. Clusters are grouped into three groups depending on changes in signal. (**E**) Genome browser snapshots of genes in Group one and genes in Group 3. TBP ChIP-seq from asynchronous and mitotic samples are shown in blue and red, respectively. (**F**) Gene Ontology term analysis of genes in Group one and Group three from (**D**). Numbers correspond to the number of genes within the group that is labeled with the specific GO term.

DOI: https://doi.org/10.7554/eLife.35621.008

The following figure supplements are available for figure 3:

**Figure supplement 1.** Synchronization for mitotic cells.
DOI: https://doi.org/10.7554/eLife.35621.009
**Figure supplement 2.** TBP ChIP-seq is highly reproducible.
DOI: https://doi.org/10.7554/eLife.35621.010

Focusing at the TSS, TBP ChIP-seq shows similar levels of enrichment at the TSS between asynchronous and mitotic cells (**Figure 3C**, **Figure 3—figure supplement 2**). Unbiased k-means clustering with k = 6 (**Figure 3D**, **Figure 3—figure supplement 2F**) shows three main groups. Group 1 (higher in mitosis) includes the first two clusters, group 2 consists of genes that display no change, and group 3 (higher in asynchronous) is composed of the last three clusters (**Figure 3D**). Visualization

of the data for specific genes in group 1 and 3 are shown in *Figure 3E*. Gene ontology (GO) term analysis of genes within each group shows that groups 1 and 3 are enriched for similar classes of genes, primarily ones involved in the cell cycle (*Figure 3F*). This analysis suggests that the differences in TBP binding that we observe by ChIP-seq largely reflect the cyclical expression of genes during the cell cycle. Despite the variations in TBP levels, most active genes retain TBP binding in mitosis, suggesting a global bookmarking activity at promoters by TBP.

## TBP recruits a subset of RNA polymerase II molecules to mitotic chromosomes

To determine the functional effect of mitotic TBP binding, we endogenously knocked-in the plant-specific minimal auxin-inducible degron (mAID) (*Holland et al., 2012*; *Nishimura et al., 2009*) at the TBP locus and generated homozygous mAID-TBP fusion proteins (C94 cell line). This fusion enables acute, auxin-inducible degradation of the mAID-tagged TBP via the ubiquitin degradation pathway (*Figure 4A*). Within 6 hr of IAA (auxin) treatment, only about 3% of mAID-TBP molecules remains detectable by bulk protein analysis with Western blot (*Figure 4B*) and by immunofluorescence using α-TBP antibody (*Figure 4C*).

One potential function of TBP binding in mitosis is to maintain promoter accessibility and chromatin architecture. To test this hypothesis, we performed ATAC-seq analysis on asynchronous and synchronized mESCs with and without TBP degradation in three biological replicates (*Figure 4—figure supplement 1*). Analyzing reads under 100 bp shows that in both asynchronous and mitotic mESCs, the accessibility at the TSS remains largely maintained even after TBP degradation (*Figure 4D*, top). To further quantify changes, we measured the $\log_2$ of untreated vs TBP-degraded samples, and averaged the levels at the TSS of all genes (*Figure 4D*, bottom). The minor reductions in levels suggest that chromatin accessibility at the promoter may be determined by underlying sequences rather than TBP binding. Furthermore, TBP degradation has little or no effect on accessibility at enhancers and other TF binding sites, or at CTCF sites (*Figure 4—figure supplement 2*). For mono-nucleosomal reads (180–250 bp), we observe a consistent decrease in read counts surrounding the TSS in asynchronous population after TBP degradation, but the overall pattern of chromatin architecture remains the same (*Figure 4E*). The effect of TBP degradation in mitosis is largely confined to the −1 nucleosome, showing a decrease at this location compared to neighboring nucleosomes (*Figure 4E*, bottom). Consistent with the short reads analysis, these results suggest that the chromatin architecture of promoters is largely governed by *cis*-elements (DNA sequence and shape) and/or chromatin-modifying factors.

To test the role of TBP in recruiting Pol II to mitotic chromosomes, we endogenously knocked-in HaloTag at the Rpb1 locus, the catalytic subunit of RNA Polymerase II, in the mAID-TBP (C94) cells. We successfully produced heterozygous knock-in of Halo-Pol2 II, as shown by Western blot analysis (*Figure 4F*), and the tagged protein localizes within the nucleus as expected (*Figure 4G*). Furthermore, ChIP-qPCR analysis using antibodies against Pol II (targeting total Pol II) and Flag (located between Halo and the N-terminal of Rpb1, targeting only Halo-Pol II) showed that the tagged Pol II binds to regions where the endogenous Pol II binds (*Figure 4—figure supplement 3*), confirming that the tagged Pol II remains functional.

To examine the dynamics of Pol II in interphase and mitotic cells, we performed spaSPT with subsequent model fitting using Spot-on and extracted the fraction of bound Pol II molecules (*Hansen et al., 2018*). As with TBP, the 2-state kinetic model fitted the Pol II data poorly (*Figure 5—figure supplements 1*). Fitting a 3-state kinetic model, we extracted the fraction of bound Halo-Pol II molecules for all the tested conditions (*Figure 5A,B*). During interphase, an average of 29.3% (S.D. 3.3%) of Pol II molecules are 'bound' (*Figure 5A*). This 'bound' population consists of both specific binding and non-specific chromatin interactions, as HaloTag by itself displays non-specific associations of 8.5% (S.D. 3.1%) on average (*Figure 5A*). To assess the specificity of the bound population, we treated the cells with either Flavopiridol, an inhibitor of Pol II promoter escape, or Triptolide, an inhibitor of Pol II initiation. After treatment with Flavopirodol or Triptolide, the bound Pol II fraction decreases to 18.0% (S.D. 5.4%) and 15.0% (S.D. 3.8%), respectively. TBP degradation in interphase cells also leads to a marked decrease in bound Pol II, to 21.9% (S.D. 4.7%), confirming the role of TBP in recruiting Pol II to chromatin. Combining TBP degradation with either drug treatment did not result in any further significant change in the fraction bound compared to drug treatments alone, suggesting that the majority of Pol II recruited by TBP is already sensitive to drug inhibition.

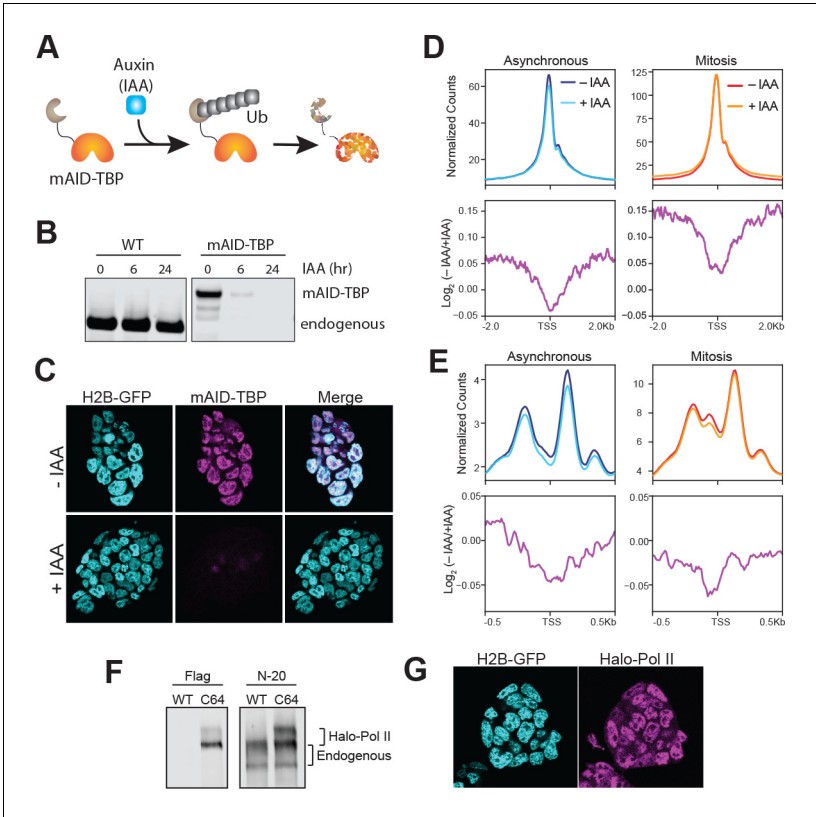

**Figure 4.** Drug-inducible degradation of endogenous TBP. (**A**) Schematic for mechanism of auxin-inducible degradation. Ub, ubiquitin. (**B**) Western blot analysis for time-course of Auxin (IAA)-dependent degradation of WT cells or cells with the endogenous knock-in of the mAID to the TBP locus. (**C**) Immunofluorescence using $\alpha$-TBP of cells with endogenous mAID-TBP and stably expressing H2B-GFP without IAA (top) or after 6 hr of IAA treatment (bottom). (**D**) ATAC-seq analysis of reads under 100 bp. Global average signal (top) surrounding the TSSs of all genes were generated for asynchronous samples (left) that were untreated (– IAA; dark blue) and TBP-degraded (+IAA; light blue), and for mitotic samples (right) that were untreated (– IAA; red) and TBP-degraded (+IAA; orange). Bottom, the corresponding $\log_2$ ratio of (– IAA/+IAA) plotted in 4 kb surrounding the TSS. (**E**) ATAC-seq analysis of 180–250 bp reads (mono-nucleosome-sized). The scheme is the same as in (**D**), with the exception that the plots are centered 1 kb surrounding the TSS. (**F**) Western blot analysis of wild-type (WT) ES cells or Halo-Pol II knock-in (C64) using $\alpha$-Flag to detect the Halo-Flag knock-in and $\alpha$-N-20, an antibody against the N-terminal of Pol II large subunit, to detect all Pol II levels. The unphosphorylated and phosphorylated bands for endogenous and Halo-Pol II are marked. (**G**) Live imaging of Halo-Pol II showing nuclear localization as marked with H2B-GFP.
DOI: https://doi.org/10.7554/eLife.35621.011

The following figure supplements are available for figure 4:

**Figure supplement 1.** ATAC-seq replicates.
DOI: https://doi.org/10.7554/eLife.35621.012

**Figure supplement 2.** ATAC-seq at enhancer and CTCF sites.
DOI: https://doi.org/10.7554/eLife.35621.013

**Figure supplement 3.** Halo-Pol II binds to promoters of active genes.
DOI: https://doi.org/10.7554/eLife.35621.014

In mitotic cells, the fraction of bound Pol II molecules decreases to an average of 15.4% (S.D. 3.2%) (*Figure 5B*), about half of the fraction of bound Pol II in interphase. After treatment with Flavo-piridol or Triptolide, the bound population decreases significantly to 11.5% (S.D. 3.1%) and 12.1% (S.D. 3.0%), respectively, which suggests that a small fraction of Pol II molecules may be actively engaged during mitosis. Furthermore, TBP degradation also leads to a significant decrease in bound Pol II molecules, to an average 11.8% (S.D. 2.9%) compared to untreated mitotic cells. This suggests a model wherein TBP recruits a small but significant subpopulation of Pol II to mitotic chromosomes.

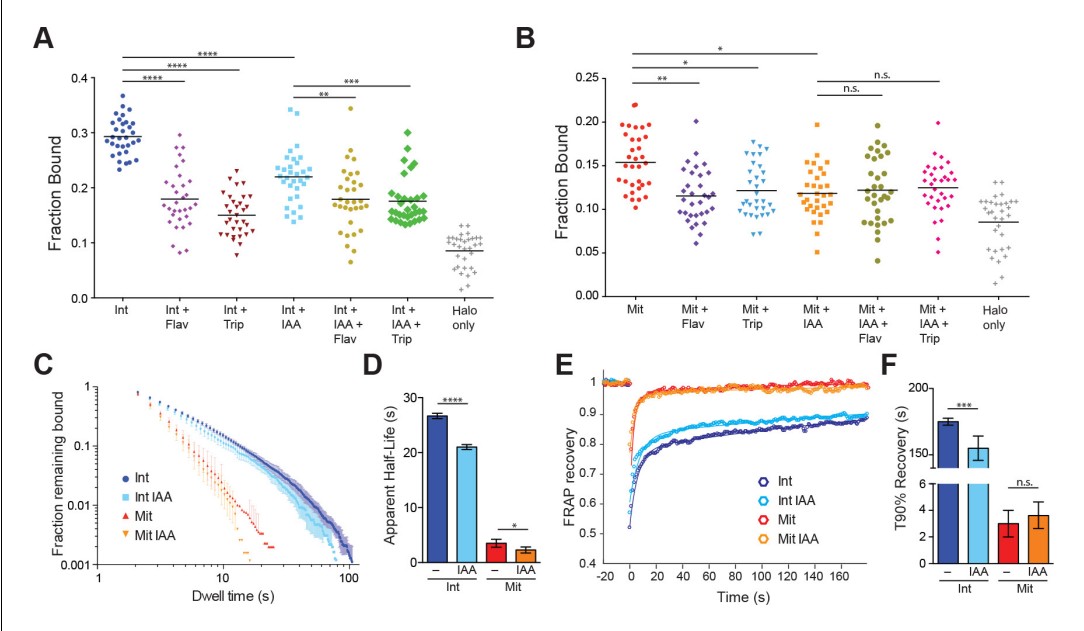

**Figure 5.** TBP-dependent dynamics of Pol II in live cells. (A) Cells were labeled with PA-JF646 and individual molecules were tracked over 25,000 frames. Scatter plot showing the fraction bound (extracted from model fitting, see Materials and methods), for each individual interphase (Int) cell, and interphase cells treated with various drugs. Flav, Flavopiridol. Trip, Triptolide, IAA, Indole Acetic Acid (TBP-degradation). Halo only corresponds to cells stably expressing the HaloTag, showing non-specific levels of 'bound' molecules. n = 32 cells over four biological replicates. Black line represents mean fraction bound. (B) Scatter plot as in (A), but for mitotic cells. Halo-only sample in (A) is replotted for direct comparison. n = 32 over four biological replicates. (C) Dwell time histogram of the fraction of endogenously-tagged Halo-Pol II molecules remaining bound for interphase (blues) and mitotic (reds) cells either with or without IAA treatment, as indicated in legend. (D) Quantification of the apparent half-life (see Materials and methods) in seconds of Halo-Pol II in interphase (blues) and mitotic (reds) cells. n = 30 cells. (E) Quantification of fluorescence recovery at the bleach spot for Halo-Pol II in interphase (blues) and mitosis (reds), either with or without IAA treatment. n = 30 cells. (F) From (E), the average time to reach 90% recovery for Halo-Pol II in interphase (blues) or mitosis (reds), either with or without IAA treatment. Data are represented as mean ± SEM. *p-value<0.05, **p-value<0.01. ***p-value<0.001, ****p-value<0.00001, n.s., not significant.

DOI: https://doi.org/10.7554/eLife.35621.015

The following figure supplement is available for figure 5:

**Figure supplement 1.** Model-fitting for Halo-Pol II in interphase (left) and mitotic (right) cells.

DOI: https://doi.org/10.7554/eLife.35621.016

As with interphase cells, combining TBP degradation with either drug treatment also did not result in a significant change in bound Pol II. Taken together, these results suggest that a small population of active Pol II binds specifically to mitotic chromosomes that is recruited by TBP.

To determine the dynamics of Pol II binding to mitotic chromosomes, we performed SPT with long exposure times (500 ms) and measured the dwell times for Halo-Pol II in interphase and mitosis with and without TBP (*Figure 5B*). Pol II binding to DNA involves complex steps, from transient binding due to promoter association and abortive initiation, to the stable binding during elongation, and to the as yet undefined mechanisms for termination. Therefore, modeling the binding of Pol II to DNA using a single residence time likely over-simplifies the complex dynamics of Pol II. Therefore, instead of extracting a single residence time for Pol II after model fitting, we calculated the apparent half-life of the bound population. During interphase, the apparent half-life of Pol II is 26.7 s (*Figure 5C*). TBP degradation leads to a modest decrease in apparent half-life, to 21.0 s, which is consistent with the prominent role of TBP in transient Pol II promoter association. During mitosis, the apparent half-life of Pol II binding decreases to 3.5 s. This order of magnitude decrease suggests that the primary mechanism that stabilizes Pol II binding, active elongation, becomes dramatically decreased during mitosis. Furthermore, after TBP degradation during mitosis, the apparent half-life decreases from 3.5 to 2.3 s, consistent with the role of TBP in recruiting transiently-binding Pol II to mitotic chromosomes.

To cross-validate the SPT residence time analysis, we performed FRAP analysis on Halo-Pol II during interphase and mitosis, with and without TBP degradation. We plotted the FRAP recovery over time (*Figure 5D*) and measured the time it takes to reach 90% recovery (T90%) (*Figure 5E*). Halo-Pol II takes 179 s to reach 90% recovery in interphase cells, whereas in mitosis, it takes 3 s. Upon TBP degradation in interphase cells, T90% decreases to 150 s, consistent with the decreased apparent half-life as measured by SPT. TBP degradation in mitotic cells shows little change in the recovery curve relative to untreated mitotic cells, further confirming our SPT residence time analysis. Taken together with the SPT results, these findings suggest that TBP affects the more transient promoter association of Pol II during interphase, and potentially directs Pol II binding to mitotic chromosomes.

## Mitotic TBP binding affects transcriptional reactivation rate

To test the hypothesis that TBP recruitment of Pol II to mitotic chromosomes helps prime active genes for efficient reactivation following mitosis, we performed two orthogonal experiments: one based on imaging and another on sequencing newly transcribed RNA products as a time course after mitosis. Using spaSPT and Spot-On to measure fraction of bound Pol II, we marked Halo-Pol II C64 cells at the metaphase stage of mitosis and imaged the same cells every 15 min until 60 min after first staging. Repeated imaging of interphase cells with and without TBP degradation showed no change in bound Pol II over time (*Figure 6A*). In contrast, the fraction of bound Pol II molecules in mitotic cells steadily increased over time, consistent with increasing activation of transcription following mitosis (*Figure 6B*). Depleting TBP during mitosis revealed that the fraction of bound Pol II remained flat over time (*Figure 6B*), suggesting that TBP is required for re-establishing the transcription program following mitosis.

Second, we extracted the nascent chromatin-associated RNA (chr-RNA) of asynchronous (A), mitotic cells (M), and 30 and 60 min after release from mitotic arrest (M30 and M60, respectively) in two replicates (*Figure 6C*, *Figure 6—figure supplements 1* and *2*). A snapshot of the spike-in-normalized mapped reads on a housekeeping (*Gapdh*) and an ES cell-specific gene (*Nanog*) is shown on *Figure 6D*. The high proportion of sequenced reads in intronic regions in asynchronous samples (*Figure 6D*, *Figure 6—figure supplement 1*) confirms enrichment for newly synthesized RNAs. To assess transcriptional activity through mitosis on a global scale, we averaged the normalized signal in a 4 kb region surrounding the TSS for all genes in each sample (*Figure 6E*). Overall levels decrease to noise levels in mitotic samples and remain low 30 min after release. However, by 60 min after release, RNA levels have increased to the levels in asynchronous samples.

To examine the role of TBP in promoting efficient reactivation, we degraded TBP in asynchronous (A) and in mitotic cells (M) (*Figure 6C*, *Figure 6—figure supplement 1*) and sequenced the chr-RNA and spike-in controls in a strand-specific manner (*Figure 6—figure supplements 1* and *2*). A snapshot of the mapped reads of TBP-degraded samples on *Gapdh* and *Nanog* is also shown on *Figure 6D*. We still observe high levels of intronic reads in asynchronous (A) samples, despite near complete degradation of TBP. Our results are reminiscent of a previous study that has observed Pol II transcription in the absence of TBP in mouse blastocyst cells (*Martianov et al., 2002*). However, we observed a marked decrease in transcription reactivation in M60 samples following TBP degradation in mitosis (*Figure 6D*). This decrease occurs globally as most genes show decreased nascent chr-RNA levels (*Figure 6F*), suggesting a specific role for TBP in transcriptional reactivation following mitosis. Intriguingly, we see that TBP degradation has an effect on tRNA (Pol III genes) but not on rRNA (Pol I genes) expression (*Figure 6—figure supplement 3*). Although this result may suggest differential roles for TBP among the three different polymerases, more stringent nascent RNA analysis will be needed to further inform this line of research.

To examine the kinetics of transcriptional reactivation following mitosis, we calculated the $\log_2$ ratio of reads in the 30- and 60 min conditions relative to mitotically arrested samoles (M30/M and M60/M, respectively), for both untreated and TBP-degraded samples. In this way, we can observe the change in RNA levels relative to mitotic cells as a function of time after release from mitosis. Globally, the untreated M30/M sample shows no overall change in RNA levels whereas M60/M samples show massive increase in both upstream and downstream transcription at the TSS (*Figure 7—figure supplement 1*). In contrast, TBP-degraded samples show delayed transcription levels evident in the M60/M sample (*Figure 7—figure supplement 1*). To determine if these changes are driven by differences in TBP levels, we measured the average change in RNA levels when genes are clustered by the three groups as determined by TBP ChIP-seq k-means clustering shown in *Figure 3D*. We

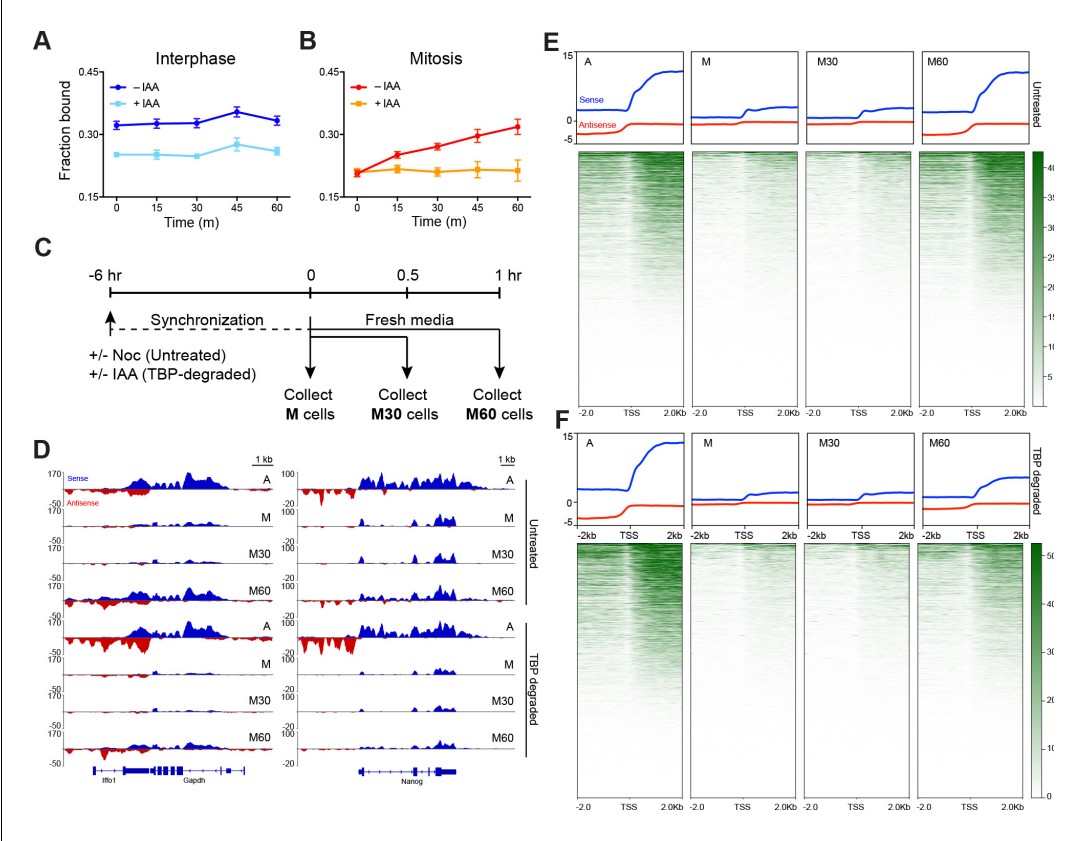

**Figure 6.** TBP recruits Pol II during mitosis. (**A**) The fraction of bound Halo-Pol II molecules as a function of time (t = 0, release from DMSO/IAA treatment) under no IAA (WT conditions) or after 6 hr of IAA treatment (teal) for interphase cells. (**B**).Same as (**A**), but for mitotic cells. (**C**) Schematic of time course regimen for extracting chromatin-associated nascent RNA. Asynchronous cells are treated with DMSO or IAA for 6 hr prior to chromatin-associated nascent RNA extraction (not depicted). For other samples, cells are synchronized with Nocodazole for 6 hr, and treated with DMSO (untreated) or IAA (TBP-degraded) during Nocodazole synchronization. For mitotic samples (M), cells are immediately collected. For time course after mitosis, synchronized M cells (untreated or TBP-degraded) are replaced into fresh media to release from mitotic arrest and TBP degradation and are collected either after 30 min (M30) or 60 min (M60) following fresh media resuspension. (**D**) Extracted chromatin-associated nascent RNA were sequenced in strand-specific manner, and the sense (blue) and anti-sense (red) reads are plotted for Gapdh (left) and Nanog (right) loci for all corresponding samples. (**E**) Genome-wide average plots for all TSS and surrounding regions for sense (blue) and anti-sense (red) reads for each indicated untreated sample, and the corresponding heatmaps for all reads (bottom). (**F**) Same as E but for TBP-degraded samples.
DOI: https://doi.org/10.7554/eLife.35621.017

The following figure supplements are available for figure 6:

**Figure supplement 1.** Nascent chr-RNA-seq is highly reproducible.
DOI: https://doi.org/10.7554/eLife.35621.018
**Figure supplement 2.** Nascent chr-RNA-seq is highly reproducible genome-wide.
DOI: https://doi.org/10.7554/eLife.35621.019
**Figure supplement 3.** Nascent chr-RNA-seq profiles for rRNA (top) and tRNA (bottom) genes for each sample.
DOI: https://doi.org/10.7554/eLife.35621.020

observed no effect between the three groups (*Figure 7—figure supplement 2*), further suggesting that the minor changes we observed by TBP ChIP-seq are likely due to changes in cyclical gene expression. We next performed unbiased k-means clustering on the untreated M30/M data with k = 3. Cluster 1 includes 5504 genes that show an increase in transcription whereas cluster 3 includes 6693 genes that show a decrease in transcription relative to mitotic cells. The remaining genes (cluster 2) show no change in transcription (*Figure 7A,B*). Ordering the M60/M data using the same three clusters shows that cluster one increases in transcription the earliest and the fastest whereas clusters 2 and 3 lag behind. We then ordered the M30/M and M60/M data from TBP-degraded samples (*Figure 7A,B*). This analysis shows that the early changes in transcription seen in untreated samples

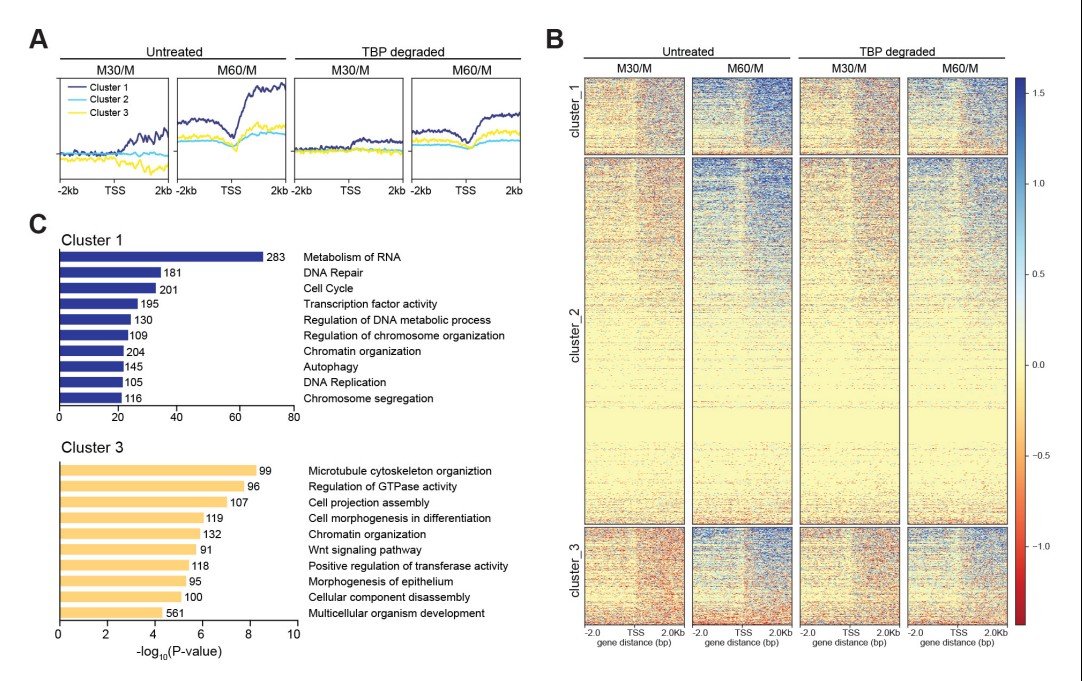

**Figure 7.** TBP promotes efficient reaction of transcription following mitosis. (**A**) The $\log_2$ ratio of M30 and M60 reads relative to M reads were calculated for untreated and TBP-degraded samples (M30/M and M60/M, respectively). The M30/M samples were clustered using k-means clustering with k = 3, and the rest of the samples are ordered by this clustering, and average plots surrounding the TSS are shown for each cluster. (**B**) Heat map analysis of each cluster for each sample in (**A**). (**C**) GO term analysis for genes present in cluster 1 (top) and cluster 3 (bottom). Numbers correspond to the number of genes within the custer that is labeled with the specific GO term.

DOI: https://doi.org/10.7554/eLife.35621.021

The following figure supplements are available for figure 7:

**Figure supplement 1.** TBP degradation affects global reactivation following mitosis.

DOI: https://doi.org/10.7554/eLife.35621.022

**Figure supplement 2.** The $\log_2$ ratio of M30 and M60 reads relative to M reads in untreated and TBP-degraded samples were grouped according to k = 3 clustering as in *Figure 3D*.

DOI: https://doi.org/10.7554/eLife.35621.023

are dampened in TBP-degraded samples with the biggest effect on cluster 1. Using GO term analysis, cluster one is enriched for genes involved in metabolism of RNA, cell cycle, and transcription factor activity (*Figure 7C*). These genes are also generally highly expressed in the asynchronous population (*Figure 7—figure supplement 2*), suggesting that these genes represent the global transcription program of the cells. In contrast, cluster three is enriched for genes involved in basic cellular processes such as microtubule cytoskeleton organization and regulation of GTPase activity, but also for genes involved in differentiation and development such as cell morphogenesis in differentiation, and morphogenesis of epithelium (*Figure 7C*). Furthermore, these genes tend to be less expressed than cluster one in the asynchronous population (*Figure 7—figure supplement 1*). Taken together, these data suggest that TBP promotes efficient reactivation of the global transcription program through cell division.

## Discussion

Although different mechanisms exist to ensure faithful transmission of transcription programs from mother to daughter cells after mitosis, the precise role of transcription factors in this process has remained elusive. In this study, we have shown that unlike most classical transcription factors that bind with rapid dynamics, the endogenous TBP maintains stable binding at the TSS of active genes in mitotic mESCs. Remarkably, such stable TBP binding leads to recruitment of Pol II to mitotic

chromosomes, despite a global decrease in transcriptional activity. Rather, as cells exit mitosis, TBP promotes efficient reactivation of transcription globally by priming the promoters of active genes. Taken together, TBP acts as a stable bookmarker of active promoters throughout mitosis and promotes transcriptional fidelity from mother to daughter mESC.

Previous biochemical and structural data have generated static views of molecular processes, such as the formation of stable complexes or binding of factors to DNA. These static views have propagated the sense of stability. For instance, a ChIP signal for TFs has generally been viewed as a binary classification of specific loci: bound or unbound. The advent of live-cell single-molecule imaging has revolutionized our understanding of basic molecular processes, particularly, how dynamic these processes are. Many TFs examined by live-cell single-molecule imaging have shown much shorter residence times on DNA than previously expected from biochemical data. For instance, Bicoid binds to the Drosophila genome for about 100 ms (*Mir et al., 2017*), p53 has an average residence time of 3–6 s (*Loffreda et al., 2017*), and Sox2 binds to specific loci in mESCs for about 12 s on average (*Chen et al., 2014*; *Teves et al., 2016*). We now have a much more dynamic view of repeated and rapid binding/unbinding events that ultimately affect transcriptional output. In contrast to most sequence-specific TFs, TBP seems unusual by binding to loci on the order of minutes. Although still more dynamic than previous biochemical data had suggested, this more stable mode of TBP binding may serve a specific function unique to TBP among TFs, perhaps as a more stable landing pad for the dynamic binding of various other factors that make up the transcriptional machinery. For instance, binding of both TFIIA and TFIIB are highly dependent on prior TFIID/TBP binding, and TFIIB has been shown to bind to TBP-bound promoter DNA in a highly transient manner through in vitro single molecule imaging (*Zhang et al., 2016*). Despite the general silencing of transcription during mitosis, this relatively stable binding of TBP is maintained. Here, we have shown that TBP recruits Pol II to mitotic chromosomes and promotes efficient reactivation of transcription following mitosis, but such stable binding in mitosis may also serve other functions. Another potential function may be to promote a decondensed state of chromatin at the TSS through interactions with phosphatase PP2A, which counteracts chromatin condensation that occurs in mitosis (*Xing et al., 2008*). In this way, TBP may promote transcriptional fidelity through mitosis in multiple ways.

A previous study has shown that mouse blastocyst cells with homozygous deletion of the TBP locus still contain Pol II in active phosphorylation states, suggestive of active transcription in the absence of TBP (*Martianov et al., 2002*). Furthermore, the authors found that transcription from Pol I and Pol III are dramatically reduced without TBP, suggesting a differential dependency of the various polymerases on TBP. In this study, we have used a rapid drug-inducible depletion of TBP and measured the newly transcribed RNA after depletion. We have confirmed that Pol II transcription occurs even in the absence of TBP (*Figure 6D*), but that Pol III transcription is dramatically decreased (*Figure 6—figure supplement 3*). How does Pol II transcription occur in the absence of TBP? Several mechanisms may potentially answer this question. First, TBP-like factor (TLF, also known as TBP-related factor 2 – TRF2), a closely related protein to TBP containing a similar saddle-like core domain for DNA binding, has been proposed to substitute for TBP in certain specialized cases (*Akhtar and Veenstra, 2011*; *Dantonel et al., 1999*). However, unlike the constant expression of TBP, previous studies have shown that TLF is not expressed in early mouse embryogenesis (*Martianov et al., 2001*, *2002*). A second potential mechanism for TBP-independent transcription is that the SAGA complex, which shares many of the TBP-associated factors found in TFIID (*Baptista et al., 2017*; *Brand et al., 1999*; *Wieczorek et al., 1998*), may be able to compensate for the absence of TBP once transcription is on going as in interphase. A third potential mechanism, and not mutually exclusive with the first two, is that independent mechanisms exist for the initiation of transcription and the subsequent re-initiation. Our finding that TBP depletion strongly inhibits activation of genes following mitosis is consistent with this theory. That is, the first transcription event following mitosis requires TBP, providing a rationale for TBP mitotic bookmarking, whereas subsequent transcription events throughout interphase may occur even in the absence of TBP.

Our observation that TBP recruits Pol II to mitotic chromosomes may seem surprising, given that historically, transcription has been shown to be globally decreased during mitosis (*Prescott and Bender, 1962*). Recent studies, however, are beginning to shed some nuance to this historical adage. For instance, several groups have shown that transcription transiently spikes at the later stages of mitosis, followed by an adjustment back to normal interphase levels (*Hsiung et al., 2016*;

*Vaňková Hausnerová and Lanctôt, 2017*). One potential mechanism for such a spike in transcriptional activity could derive from our observed recruitment of Pol II by TBP to mitotic chromosomes. Combined with the lack of crosstalk between enhancers and promoters during mitosis (*Hsiung et al., 2015*), it is possible that Pol II recruitment results in promiscuous transcriptional activation of most accessible promoters immediately following mitosis. Adding more nuance, several groups have recently shown that the global transcription pattern is retained albeit at low levels during mitosis (*Liang et al., 2015*; *Palozola et al., 2017*). It is possible that the small fraction of Pol II that we observe to be recruited to mitotic chromosomes is transcriptionally active, especially since these bound molecules are also sensitive to drug inhibition (*Figure 5B*). If so, this activity must be quite transient, as we observe no detectable Pol II molecules that bind to mitotic chromosomes for longer than 30 s by slow tracking SPT (*Figure 5C*). In contrast, we observe much longer residence times of Pol II during interphase when transcription is highly active. Indeed, less than 1% of mitotically bound Pol II molecules show binding for greater than 10 s (*Figure 5B*), which may reflect the low transcriptional activity during mitosis as previously reported (*Liang et al., 2015*; *Palozola et al., 2017*). What does become evident, however, is the gradual increase in bound Pol II concomitant with increase in newly transcribed RNA as cells progress from mitosis, and the important role of TBP in this process (*Figure 6*). Regardless of the transcriptional activity of Pol II during mitosis per se, TBP promotes efficient reactivation of global transcription following mitosis.

## Materials and methods

### Key resources table

| Reagent type (species) or resource | Designation | Source or reference | Identifiers | Additional information |
|---|---|---|---|---|
| Genetic reagent (mouse cell line) | JM8.N4 mouse ES cells | KOMP repository | RRID: CVCL_J962 | Parental cell line used for all genetic manipulations |
| Antibody | TBP | Abcam | Abcam #ab51841; RRID:AB_945758 | 1:250 for Western; 10 μg for ChIP |
| Antibody | N-20 RNA Polymerase II | Santa Cruz | Santa Cruz #sc-899; RRID:AB_632359 | 1:250 for Western |
| Antibody | 8WG16 Pol II | Santa Cruz | RRID:AB_785522 | 1:250 for Western |
| Antibody | Flag | Sigma Aldrich | Sigma-Aldrich Cat# F3165, RRID:AB_259529 | 1:5000 for Western |
| Cell line (mouse) | Halo-TBP KI C41 | This paper | Halo-TBP KI C41 | Endogneous knock-in of HaloTag to N-ternimal of TBP in JM8.N4 cells |
| Cell line (mouse) | mAID-TBP KI C94 | This paper | mAID-TBP KI C94 | Endogenous knock-in of the minimal auxin inducible degron to N-terminal of TBP in JM8.N4 cells |
| Cell line (mouse) | Halo-Pol II C64 | This paper | Halo-Pol II C64 | Endogenous knock-in of the HaloTag to C-terminal of Rbp1, largest subunit of Pol II, in mAID-TBP C94 cells |
| Commercial assay or kit | Nextera DNA Library Preparation Kit | Illumina | FC-121–1030 | ATAC-seq reagent for Tn5 transposition |
| Software, algorithm | Spot-On | doi: 10.7554/eLife.25776 | | Spot-on software used to analyze spaSPT data |

### Cell culture

For all experiments, we used the mouse ES cell line JM8.N4 (RRID: CVCL_J962) obtained from the KOMP repository (https://www.komp.org/pdf.php?cloneID=8669) and tested negative for mycoplasma. ES cells were cultured on gelatin-coated plates in ESC media Knockout D-MEM (Invitrogen, Waltham, MA) with 15% FBS, 0.1 mM MEMnon-essential amino acids, 2 mM GlutaMAX, 0.1 mM 2-

mercaptoethanol (Sigma) and 1000 units/ml of ESGRO (Chem- icon). ES cells are fed daily and passaged every two days by trypsinization. Mitotic cells were synchronized by adding 100 ng/mL of Nocodazole for 6 hr followed by shake-off. To assess the purity of synchronization, cells collected from shake off were placed on gelatin-coated glass bottom microwell dishes (MatTek #P35G-1.5–14 C) and imaged for H2B-GFP using epifluorescence microscopy. Mitotic cells were counted by visual inspection. For endogenously-tagged mAID-TBP cells, TBP degradation was performed by addition of indole-3-acetic acid (IAA) at 500 µM final concentration for 4–6 hr. Flavopiridol and triptolide treatments were performed at 1 µM final concentration for 30–60 min.

## Cas9-mediated endogenous knock-ins

Cas9-mediated endogenous knock-ins were performed as previously described (*Teves et al., 2016*). Briefly, mESCs were transfected with 0.5 µg of the Cas9 vector (containing the specific guide RNA and the Venus coding sequence) and with 1 µg of the donor repair vector using Lipofectamine 3000. The transfected cells were sorted for Venus expressing cells the next day. Sorted cells were plated and grown on gelatin-coated plates under dilute conditions to obtain individual clones. After one week, individual clones were isolated and were used for direct cell lysis PCR using Viagen DirectPCR solution. PCR positive clones were further grown on gelatin-coated plates and cell lysates were obtained for Western blot analysis. Homozygous and heterozygous Halo-tagged TBP cell lines were generated, one homozygous of which was further tested by teratoma assay and was performed by Applied Stem Cell. We also generated homozygous mAID-TBP cell lines, one of which (C94) was further used to endogenously knock-in HaloTag to the Rpb1 locus, the largest subunit of RNA Polymerase II (C64 cell line). All primers and guide RNAs used to generate knock-ins are listed in *Supplementary file 1*. To verify protein tagging, cell lysates were analyzed by Western blotting and probed for TBP (Abcam #ab51841; RRID:AB_945758), N20 Pol II antibody (Santa Cruz #sc-899; RRID:AB_632359), 8WG16 Pol II antibody (RRID:AB_785522), or Flag antibody (Sigma-Aldrich Cat# F3165, RRID:AB_259529).

## Live cell and fixed sample imaging

Live-cell imaging experiments including FRAP were performed on cells grown on gelatin-coated glass bottom microwell dishes (MatTek #P35G-1.5–14 C) and labeled with the Halo-ligand dye JF549 at 100 nM concentration for 30 min. To remove unbound ligand, cells were washed 3x with fresh media for 5 min each. Epi-fluorescence time lapse imaging was performed on Nikon Biostation IM-Q equipped with a 40x/0.8 NA objective, temperature, humidity and CO2 control, and an external mercury illuminator. Images were collected every 2 min for 12 hr. Confocal live-cell imaging was performed using a Zeiss LSM 710 confocal microscope equipped with temperature and CO2 control. For fixed imaging, labeled cells were fixed with 4% PFA for 10 min in room temperature and were washed with 1x PBS prior to imaging. Standard immunofluorescence was performed on labeled cells using 4% PFA. Quantification of chromosome enrichment was performed using Fiji.

## FRAP

FRAP was performed on Zeiss LSM 710 confocal microscope with a 40x/1.3 NA oil-immersion objective and a 561 nm laser as previously described (*Teves et al., 2016*). Bleaching was performed using 100% laser power and images were collected at 1 Hz for the indicated time. FRAP data analysis was performed as previously described (*Hansen et al., 2017*). For each cell line, we collected 10 cells for technical replicates in one experiment, which was repeated for a total of three biological replicates (30 cells total).

## Single molecule imaging – Slow tracking

Cells were labeled with JF549 at 10 pM for 30 min, and washed 3x with fresh media for 5 min each to remove unbound ligand. Cells were imaged in ESC media without phenol-red. A total of 600 frames were collected for imaging experiments at 500 ms frame rate (2 Hz) and were repeated 3x for biological replicates, with each experiment consisting of 10 cells for technical replicates each. Data are represented as mean over experimental replicates (30 cells total) ± SEM. Imaging experiments were conducted as described previously (*Teves et al., 2016*) on a custom-built Nikon TI microscope equipped with a 100x/NA 1.49 oil-immersion TIRF objective (Nikon apochromat CFI

Apo SR TIRF 100x Oil), EM-CCD camera (Andor iXon Ultra 897), a perfect focusing system (Nikon) and a motorized mirror to achieve HiLo-illumination (*Tokunaga et al., 2008*). Bound molecules were identified using SLIMfast as previously described (*Teves et al., 2016*). The length of each bound trajectory, corresponding to the time before unbinding or photobleaching, was determined and used to generate a survival curve (fraction still bound) as a function of time. A two-exponential function was then fitted to the survival curve, and the residence time was determined as previously described (*Teves et al., 2016*). The apparent half-life in seconds of bound Halo-Pol II was calculated by dividing the natural log of 2 with the photobleaching-corrected $k_{off}$ ($\ln2/k_{off}$) extracted from the two-exponential function fitting. Photobleaching correction is performed by subtracting the apparent $k_{off}$ of H2B-Halo from the apparent $k_{off}$ of Pol II samples as previously described (*Hansen et al., 2017*; *Teves et al., 2016*).

## Single molecule imaging – Fast tracking
Cells were labeled with photo-activatable PA-JF549 or PA-JF646 (as indicated) at 25 nM for 30 min and washed 3x with fresh media for 5 min each to remove unbound ligand. Cells were imaged in ESC media without phenol-red. A total of 20,000 frames were collected for at 7.5 ms frame rate (133 Hz) and were repeated 3-4x for biological replicates, with each replicate consisting of eight cells of technical replicates. Data are represented as mean over experimental replicates (24–32 cells total) ± standard error of means. Single molecules were localized and tracked using SLIMfast, a custom-written MATLAB implementation of the MTT algorithm (*Sergé et al., 2008*), using the following algorithm settings: Localization error: $10^{-6.25}$; deflation loops: 0; Blinking (frames); 1; maximum number of competitors: 3; maximal expected diffusion constant (µm2/s): 20. The fraction of bound molecules was determined as previously described using the following fit paramerters: TimeGap, 7.477; Gaps Allowed, 1; Jumps to consider, 4; Model fit type, CDF; Localization error, 0.045 (*Hansen et al., 2017*, *2018*; *Teves et al., 2016*) using Spot-On (source code freely available at https://gitlab.com/tjian-darzacq-lab/spot-on-matlab). The general statistics for image analysis (number of detections, number of total trajectories, and number of trajectories $\geq$ 3) for all spaSPT data are listed in *Supplementary file 2*.

## ChIP-seq
ChIP-seq was performed as described previously (*Skene and Henikoff, 2015*). Briefly, asynchronous and synchronized mitotic mESCs (JM8.N4) were cross-linked using 1% formaldehyde for 5 min at room temperature and washed with PBS before quenching with 125 mM Glycine in PBS for 5 min. Cells were scraped, collected by centrifugation, and resuspended in 200 µL of Lysis buffer (1% SDS, 10 mM EDTA, 50 mM Tris-HCl (pH 8.1), protease inhibitors) for 10 min on ice. Lysates were diluted 10-fold in cold ChIP dilution buffer (1% Triton X-100, 20 mM Tris-HCl (pH 8.1), 2 mM EDTA, 150 mM NaCl, 5 mM CaCl$_2$). MNase (5 µL of 0.2 U/µL stock) was added to the lysate and samples were incubated at 37C for 15 min. Digested was quenched by adding EDTA (final concentration of 10 mM) and EGTA (final concentration 20 mM). Samples were sonicated using Branson digital sonifier at 40 s on time at 30% power with 2.5 s on and 5 s off cycles. Lysates were cleared by centrifugation and used as input for ChIP by adding α-TBP antibody (Abcam #ab51841 RRID:AB_945758) and incubating overnight. Protein-G magnetic beads were used to immuno-capture bound fragments, which were then washed as described previously (*Skene and Henikoff, 2015*), and DNA was purified by ethanol extraction. Two replicates were performed, and each sample and replicate was sequenced using one lane of Illumina Hi-Seq 2500 for 50 bp paired-end reads. Reads were mapped on mm10 genome build using Bowtie2 with the following parameters: –no-unal –local –very-sensitive-local –no-discordant –no-mixed –contain –overlap –dovetail –phred33 –I 10 –X 2000. Peak calling was performed using MACS2 with the following parameters: –format BAMPE; -g mm (*Zhang et al., 2008*). Differential peak analysis was performed using DiffBind (*Ross-Innes et al., 2012*; *Stark and Brown, 2011*). TSS heat map and average plot analyses were performed using DeepTools suite (*Ramírez et al., 2016*). TSSs of mouse genome is generated from start site locations of all refseq genes from the UCSC genome browser (mm10). Data is deposited in GEO under the accession number GSE109964.

## Chr-RNA-seq

Chromatin associated nascent RNA was extracted as previously described (*Teves and Henikoff, 2011*). Briefly, 10 million asynchronous or mitotic cells were collected for each sample as described in *Figure 6C*. Each sample was washed with ice-cold PBS and lysed with 800 µL of Buffer A (10 mM HEPES, pH 7.9, 10 mM KCl, 1.5 mM MgCl$_2$, 0.34 M Sucrose, 10% Glycerol, 1 mM DTT, 0.1% Triton X-100, protease inhibitors) as described. Nuclei were subjected to consecutive biochemical fractionation with incubations for 15 min each with centrifugation and collection in between each fractionation. Nuclei were incubated 3 times for 15 min with Buffer B (9 mM EDTA, 20 mM EGTA, 1 mM DTT, 0.1% Triton X-100, protease inhibitors), then 2 times for 15 min each with Buffer B+ (20 mM EDTA, 20 mM EGTA, 2 mM spermine, 5 mM spermidine, 1 mM DTT, 0.1% Triton X-100, protease inhibitors). Following fractionation, RNA from the insoluble chromatin was extracted with Trizol and prepared for sequencing as previously described (*Teves and Henikoff, 2011*). Sequencing was performed on one lane of Hi-Seq 4000 with 50 bp single-end reads.

## Chr-RNA-seq analysis

The sequenced reads were mapped to either the mm10 genome build or ERCC92 build for the spike-in controls using Tophat with the following parameters: library-type = fr firststrand, b2-very-sensitive, no-coverage-search. The mapped reads were de-duplicated using the samtools rmdup function and then scaled using the scaling index calculated from the total number of reads that mapped to the ERCC92 build. The scaling equation is 1000/# of de-duplicated ERCC-mapped reads (*Supplementary file 3*). Bigwig files were generated using DeepTools suite bamCoverage function, and reads were extended for 200 bp. Intronic transcripts per million (TPM) counts for each gene were calculated using Bioconductor R, which were then inputted for principal component analysis. Scatter plots were generated using R ggplots package. TSS plots, heatmaps, and k-means clustering analyses were generated with combined replicates using DeepTools suite (*Ramírez et al., 2016*). Data is deposited in GEO under the accession number GSE109964.

## ATAC-seq

ATAC-seq was performed as previously described (*Buenrostro et al., 2013*) with the following modifications. Asynchronous cells were trypsinized and counted. Mitotic cells were collected by shake off following Nocodazole treatment. After cells were counted, a total of 10,000 cells were pelleted for each condition and washed with 50 µL of PBS. Washed cells were pelleted and immediately resuspended in transposase reaction mix (25 µL 2x TD buffer, 2.5 µL transposase, and 22.5 µL nuclease-free water). Transposition and DNA purification was performed as described (*Buenrostro et al., 2013*). For PCR amplification, we performed amplification for library preparation using 12 cycles for all samples. We performed three replicates, and each sample and replicate was sequenced using Hi-Seq 4000 using 100 bp paired end reads.

## ATAC-seq data analysis

All paired end reads were assessed using FastQC, and trimmed to 25 bp for each end using Trimmomatic with the following arguments: phred, 33; CROP, 25 (*Bolger et al., 2014*). Trimmed reads were then mapped to the mm10 genome build using Bowtie2 with the following arguments: –no-unal –local –very-sensitive-local –no-discordant –no-mixed –contain –overlap –dovetail –phred33. Optical and PCR duplicates were removed after mapping using Picard MarkDuplicates tool. Mapped reads from each sample was further subsampled to have the same proportion of reads for all samples. Bigwig files for each size class (under 100 bp, 180–250 bp) were generated using DeepTools suite bamCoverage function, with the option –MNase set for the size class 180–250 bp. TSS plots, heatmaps, and k-means clustering analyses were generated with combined replicates using DeepTools suite (*Ramírez et al., 2016*). Data is deposited in GEO under the accession number GSE109964.

## Acknowledgements

We thank Luke Lavis for generously providing JF dyes, Gina M Dailey for assistance with cloning, Astou Tangara and Anatalia Robles for microscopy assembly and maintenance, and Dr. Kartoosh Heydari at the Li Ka Shing Facility for flow cytometry assistance. We thank Anders Hansen, Claudia

Cattoglio, Claire Darzacq and other members of the Tjian and Darzacq labs, for insightful comments on the manuscript. This work was performed in part at the CRL Molecular Imaging Center, supported by the Gordon and Betty Moore Foundation. This work used the Vincent J Coates Genomics Sequencing Laboratory at UC Berkeley, supported by NIH S10 Instrumentation Grants 10RR029668 and S10RR027303. SST is a postdoctoral fellow of the Jane Coffin Childs Foundation. This work was supported by the California Institute of Regenerative Medicine grant LA1-08013 (XD), and by the Howard Hughes Medical Institute (003061, RT). Sequencing data has been deposited at NCBI GEO under accession code GSE109964. A preprint describing this work was first available on BioRxiv January 2018: https://www.biorxiv.org/content/early/2018/01/31/257451.

## Additional information

### Competing interests
Robert Tjian: One of the three founding funders of eLife and a member of eLife's Board of Directors. The other authors declare that no competing interests exist.

### Funding

| Funder | Grant reference number | Author |
| --- | --- | --- |
| Jane Coffin Childs Memorial Fund for Medical Research | | Sheila S Teves |
| Siebel Stem Cell Institute | | Sheila S Teves<br>Robert Tjian |
| California Institute for Regenerative Medicine | LA1-08013 | Xavier Darzacq |
| Howard Hughes Medical Institute | | Robert Tjian |

The funders had no role in study design, data collection and interpretation, or the decision to submit the work for publication.

### Author contributions
Sheila S Teves, Conceptualization, Data curation, Formal analysis, Supervision, Funding acquisition, Investigation, Visualization, Methodology, Writing—original draft, Writing—review and editing; Luye An, Aarohi Bhargava-Shah, Data curation, Investigation, Writing—review and editing; Liangqi Xie, Resources, Methodology, Writing—review and editing; Xavier Darzacq, Robert Tjian, Conceptualization, Supervision, Funding acquisition, Writing—review and editing

### Author ORCIDs
Sheila S Teves ![ORCID] http://orcid.org/0000-0002-1220-2414
Xavier Darzacq ![ORCID] http://orcid.org/0000-0003-2537-8395
Robert Tjian ![ORCID] https://orcid.org/0000-0003-0539-8217

### Decision letter and Author response
Decision letter https://doi.org/10.7554/eLife.35621.034
Author response https://doi.org/10.7554/eLife.35621.035

## Additional files

### Supplementary files
• Supplementary file 1. Primers and guide RNAs for Cas9-mediated knock-ins. Primers were ordered as custom DNA oligos from IDT in 25 nmole amounts using standard desalting methods.
DOI: https://doi.org/10.7554/eLife.35621.024

• Supplementary file 2. Statistics on spaSPT imaging analysis. For all Halo-TBP and Halo-Pol II spaSPT imaging experiments, statistics on total number of trajectories, number of trajectories greater than 3, total number of localizations, localization per frame, total number of jumps, and jumps used for modeling are reported.
DOI: https://doi.org/10.7554/eLife.35621.025

• Supplementary file 3. Scaling factors for chr-RNA-seq replicates. For each chr-RNA-seq replicate, the total number of sequenced ERCC spike-in control was determined. The scaling factor for each replicate was determined by 1000 divided by the number of sequenced ERCC spike-in control.
DOI: https://doi.org/10.7554/eLife.35621.026

• Transparent reporting form
DOI: https://doi.org/10.7554/eLife.35621.027

## Data availability

Sequencing data have been deposited in GEO under accession code GSE109964.

The following dataset was generated:

| Author(s) | Year | Dataset title | Dataset URL | Database, license, and accessibility information |
|-----------|------|---------------|-------------|--------------------------------------------------|
| Teves SS | 2018 | A stable mode of bookmarking by TBP recruits RNAPolymerase II to mitotic chromosomes | https://www.ncbi.nlm.nih.gov/geo/query/acc.cgi?acc=GSE109964 | Publicly available at the NCBI Gene Expression Omnibus (accession no: GSE109964) |

The following previously published dataset was used:

| Author(s) | Year | Dataset title | Dataset URL | Database, license, and accessibility information |
|-----------|------|---------------|-------------|--------------------------------------------------|
| Tveves SS, Tjian R | 2016 | Global accessibility of mitotic chromosomes | http://www.ncbi.nlm.nih.gov/geo/query/acc.cgi?acc=GSE85184 | Publicly available at the NCBI Gene Expression Omnibus (accession no: GSE85184) |

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
