## [Decision Letter]

Thank you for submitting your article "A stable mode of bookmarking by TBP recruits RNA Polymerase II to mitotic chromosomes" for consideration by *eLife*. Your article has been reviewed by three peer reviewers, one of whom is a member of our Board of Reviewing Editors, and the evaluation has been overseen by Aviv Regev as the Senior Editor.

The reviewers have discussed the reviews with one another and the Reviewing Editor has drafted this decision to help you prepare a revised submission.

Summary:

In their manuscript, Teves et al. report the results of a series of elegant experiments aimed at elucidating the role of TBP in transcriptional reactivation after mitosis. The authors used a suite of advanced approaches, such as epitope-tagging of endogenous TBP and Pol II, various live imaging techniques, and degron-mediated depletion of endogenous TBP. These approaches provide a superior experimental approach relative to previous published work investigating TBP action during mitosis. Key results can be summarized as follows:

1) Unlike many other TFs, TBP remains strongly bound to chromatin during mitosis.

2) TBP binds to promoters of active genes during mitosis.

3) TBP is partly required for Pol II recruitment to chromatin both in interphase and mitosis.

4) In the absence of TBP, Pol II reactivation after mitosis is impaired at a large subset of genes.

The reviewers deemed the results interesting and appreciated the novel approaches employed. However, the reviewers also identified and discussed a number of major concerns, and decided to invite resubmission of a revised paper addressing these.

Reviewers noted that the novelty of the results presented needs to be better placed into the context of the published literature. Segil et al., (1996) showed that a sub-population of TBP is visible on chromosomes of mitotic cells and that TBP and TAF12 were clearly present in their highly purified mitotic chromosome preparation. Christova et al. (2002) showed that TFIID (TBP and TAF5) remains associated with active class II gene promoters during mitosis (the authors did not cite this study). Fairley et al. (2003) showed that TBP and Brf1 remain associated with purified metaphase chromosomes (the authors did not cite this study). Xing et al. (2008) reported that TBP-PP2A mitotic complex bookmarks genes by preventing condensin action. This same team (Teves et al., 2016) showed also that TBP fractionated with mitotic chromosomes (Figure 1—figure supplement 2). All these five studies demonstrate unequivocally that TBP is bound to mitotic chromosomes independently of the methodology used without any conflicting results.

In the light of all these studies, the reviewers agreed that the current manuscript should provide more insight in order to advance the field in a significant way.

Major concerns:

1) Need for further insights about bookmarking, transcriptional memory, and transcriptional reactivation after mitosis. The entire manuscript, including the title, is written under the overall interpretation/assumption that TBP is required for 'bookmarking' of genes that were active during interphase, so that these same genes can be transcribed again after mitosis. Thus, TBP is deemed to confer a 'transcriptional memory' to preserve cellular identity after cell division. Reviewers discussed this interpretation and concluded that more data is necessary to sustain these conclusions.

The results show that TBP is required for 'reactivation' of many genes after mitosis (although many genes are still 'reactivated' without TBP), but there is no evidence that TBP is part of the initial 'bookmarking' event that confers 'transcriptional identity/memory'. In the absence of TBP, there does not seem to be any loss of 'transcriptional memory' i.e. when mouse ES cells are depleted of TBP, they still seem to express the same set of genes, and Pol II is not suddenly redirected to new sets of genes, or impeded from transcribing the genes that were expressed in the previous interphase. In other words, TBP seems to act downstream of a primary bookmarking event that defines the sets of active promoters in this cell type, and, as it would be expected for a core subunit of a general transcription factor, TBP is required for reactivation of Pol II at previously active genes.

These issues can be addressed experimentally with technologies already used in the manuscript. What would be the impact of TBP depletion on ATAC-seq results shown in Figure 1A, B? Would TBP depletion erase the 'enhancer memory' or 'nucleosomal memory' of murine ES cells? Would entire sets of active promoters be occluded upon TBP depletion while others would be 'licensed' during the course of a single TBP-less mitosis? Reviewers discussed that the bookmarking mechanism would be the mechanism that enabled global promoter nucleosome architecture to be basically the same in asynchronous versus mitotic cells sown in Figure 1B, and there is currently not proof that TBP is involved in this mechanism.

A second issue within this same framework revolves around the interpretation of the impact of TBP depletion on recruitment of Pol II during interphase and mitosis. The results show that TBP remains bound to chromatin during mitosis. In contrast, the levels of chromatin-bound Pol II are much decreased during mitosis. In both stages of the cell cycle, TBP depletion mildly reduces levels of chromatin-bound Pol II. What is not addressed in the manuscript is the mechanism by which Pol II is removed from chromatin during mitosis and how it relates to TBP action. Where in the genome is the fraction of Pol II that is bound to chromatin during mitosis? Which pool of Pol II is sensitive to TBP versus sensitive to mitotic chromosome condensation? Presumably, Pol II would be depleted from gene bodies during mitosis, indicative of a shutdown in elongation and transcription, but a fraction of it would remain bound to promoters. After all, promoter nucleosome architecture is preserved in mitosis (Figure 1B). If so, it would then be the pool of promoter-bound Pol II that is sensitive to TBP depletion? This could be answered with ChIP-seq for Pol II in asynchronous cells versus mitotic cells -/+ TBP. Another way of investigating this would be to look at the impact of TBP depletion on tagged Pol II after treatment with flavopiridol (an inhibitor of elongation) or tryptolide (an inhibitor of initiation). One possibility is that TBP depletion would preferentially affect the pool of Pol II that remains bound to chromatin (potentially promoters) after flavopiridol treatment.

Also on the same topic, the data does not support the corresponding heading (Results, subsection “TBP recruits RNA Polymerase II to mitotic chromosomes”). This same team (Teves et al., 2016) showed that TBP fractionated with mitotic chromosomes, whereas Pol II did not (Figure 1—figure supplement 2 in Teves et al., 2016). In contrast, in the present study the authors show that in the heterozygous Halo-Pol II knock-in cell-line, the bound population of Halo-Pol II is 12.1% during mitosis (Results, subsection “TBP recruits RNA Polymerase II to mitotic chromosomes”, fourth paragraph). The authors should clarify this contradiction. Moreover, the effect of TBP degradation on Pol II recruitment and binding is quite mild, it changes from 12.1% in untreated mitotic cells to about 10.8% after TBP degradation. Is this change really significant? This very minor change does not support the claim that "TBP recruits RNA Polymerase II to mitotic chromosomes". The same comment stands for “Remarkably, such stable TBP binding leads to recruitment of Pol II to mitotic chromosomes, despite the general inhibition of transcriptional activity”. The authors should tone down their overstated conclusions.

Finally, when analyzing the effects of mitotic TBP binding/depletion on the transcriptional reactivation rate (in Figure 6 or Figure 7), could the authors see any difference between Group 1, 2, 3 genes (as defined in Figure 3D)?

2) Need for further insight about which TBP-containing complex impacts on Pol II reactivation after mitosis. As TBP is involved in Pol I, Pol II, and Pol III transcription, the authors should show, which TBP-containing complex, SL1, TFIID and/or TFIIB, is playing the reactivation 'bookmarking role'. This would provide more novelty over published work.

3) Asynchronous versus mitotic cells. Reviewers did not find sufficient details and controls about how the asynchronous versus mitotic cell populations were generated. Authors should demonstrate the degree of mitotic synchronization efficiency and the number of mitotic cells in the 'asynchronous' population. Otherwise the interpretation of all the ChIP-seq results becomes questionable. Alternatively, the authors should consider FACS sorting after synchronisation as described by Kadauke et al. (2012).

---

## [Author Response]

Reviewers noted that the novelty of the results presented needs to be better placed into the context of the published literature. Segil et al., (1996) showed that a sub-population of TBP is visible on chromosomes of mitotic cells and that TBP and TAF12 were clearly present in their highly purified mitotic chromosome preparation. Christova et al. (2002) showed that TFIID (TBP and TAF5) remains associated with active class II gene promoters during mitosis (the authors did not cite this study). Fairley et al. (2003) showed that TBP and Brf1 remain associated with purified metaphase chromosomes (the authors did not cite this study). Xing et al. (2008) reported that TBP-PP2A mitotic complex bookmarks genes by preventing condensin action. This same team (Teves et al., 2016) showed also that TBP fractionated with mitotic chromosomes (Figure 1—figure supplement 2). All these five studies demonstrate unequivocally that TBP is bound to mitotic chromosomes independently of the methodology used without any conflicting results.In the light of all these studies, the reviewers agreed that the current manuscript should provide more insight in order to advance the field in a significant way.

We agree with the reviewers and we have revised the Introduction section (second paragraph) to include a discussion of these important studies characterizing TBP binding to mitotic chromosomes. As the reviewers point out in their summary above, however, our manuscript goes far beyond what previous studies have done. We believe that our ability to examine the dynamics of the endogenous TBP, and our functional analyses on TBP’s role in Pol II dynamics and activity during and after mitosis provide significant advances in the field.

Major concerns:1) Need for further insights about bookmarking, transcriptional memory, and transcriptional reactivation after mitosis. The entire manuscript, including the title, is written under the overall interpretation/assumption that TBP is required for 'bookmarking' of genes that were active during interphase, so that these same genes can be transcribed again after mitosis. Thus, TBP is deemed to confer a 'transcriptional memory' to preserve cellular identity after cell division. Reviewers discussed this interpretation and concluded that more data is necessary to sustain these conclusions.The results show that TBP is required for 'reactivation' of many genes after mitosis (although many genes are still 'reactivated' without TBP), but there is no evidence that TBP is part of the initial 'bookmarking' event that confers 'transcriptional identity/memory'. In the absence of TBP, there does not seem to be any loss of 'transcriptional memory' i.e. when mouse ES cells are depleted of TBP, they still seem to express the same set of genes, and Pol II is not suddenly redirected to new sets of genes, or impeded from transcribing the genes that were expressed in the previous interphase. In other words, TBP seems to act downstream of a primary bookmarking event that defines the sets of active promoters in this cell type, and, as it would be expected for a core subunit of a general transcription factor, TBP is required for reactivation of Pol II at previously active genes.

The reviewers raise an important issue on the whether TBP can be truly considered a bookmarking factor. Our study on TBP does not make the assumption that the observed binding of TBP during mitosis constitutes the initial bookmarking event, as defined in the traditional sense. Rather, we focus on the mechanisms of TBP stably binding to promoters during mitosis and that this binding has the functional consequence of recruiting Pol II for efficient reactivation of transcription. With discoveries showing that most TFs associate with mitotic chromosomes, the term bookmarking is evolving from an all-or-none to a continuum definition. Where TBP and the Pol II machinery fit in this continuum may remain to be decided.

To respond to this main concern further, we have broken it down by concept and addressed each one individually. Overall, we find these suggestions helpful in further understanding the mechanisms of TBP and Pol II action during and after mitosis, and we think the results support the main conclusions presented in the manuscript on the role of TBP in reactivating global transcription. However, we have toned down our interpretation that “TBP is required for bookmarking … to preserve cell identity” throughout the manuscript because as the reviewers point out, we do not have direct evidence on maintenance of cell identity per se to fully support this interpretation.

These issues can be addressed experimentally with technologies already used in the manuscript. What would be the impact of TBP depletion on ATAC-seq results shown in Figure 1A, B? Would TBP depletion erase the 'enhancer memory' or 'nucleosomal memory' of murine ES cells? Would entire sets of active promoters be occluded upon TBP depletion while others would be 'licensed' during the course of a single TBP-less mitosis? Reviewers discussed that the bookmarking mechanism would be the mechanism that enabled global promoter nucleosome architecture to be basically the same in asynchronous versus mitotic cells sown in Figure 1B, and there is currently not proof that TBP is involved in this mechanism.

We agree with the reviewers that this ATAC-seq experiment in cells degraded of TBP is an interesting experiment. We therefore performed this experiment and included the analyses in the revised manuscript (Figure 4F, G, and new Figure 4—figure supplements 1-3). In summary, TBP depletion shows some minor effects on promoter chromatin architecture, suggesting that the major determinants of promoter chromatin architecture are largely independent of TBP. We also found that most sequence specific TF binding sites remain unaffected upon TBP depletion, as would be expected (Figure 4—figure supplement 2).

A second issue within this same framework revolves around the interpretation of the impact of TBP depletion on recruitment of Pol II during interphase and mitosis. The results show that TBP remains bound to chromatin during mitosis. In contrast, the levels of chromatin-bound Pol II are much decreased during mitosis. In both stages of the cell cycle, TBP depletion mildly reduces levels of chromatin-bound Pol II. What is not addressed in the manuscript is the mechanism by which Pol II is removed from chromatin during mitosis and how it relates to TBP action. Where in the genome is the fraction of Pol II that is bound to chromatin during mitosis? Which pool of Pol II is sensitive to TBP versus sensitive to mitotic chromosome condensation? Presumably, Pol II would be depleted from gene bodies during mitosis, indicative of a shutdown in elongation and transcription, but a fraction of it would remain bound to promoters. After all, promoter nucleosome architecture is preserved in mitosis (Figure 1B). If so, it would then be the pool of promoter-bound Pol II that is sensitive to TBP depletion? This could be answered with ChIP-seq for Pol II in asynchronous cells versus mitotic cells -/+ TBP. Another way of investigating this would be to look at the impact of TBP depletion on tagged Pol II after treatment with flavopiridol (an inhibitor of elongation) or tryptolide (an inhibitor of initiation). One possibility is that TBP depletion would preferentially affect the pool of Pol II that remains bound to chromatin (potentially promoters) after flavopiridol treatment.

We agree with the reviewers that the question of which Pol II species is associating with mitotic chromosomes and how much of it is TBP-dependent are important issues to address. We had previously performed ChIP-seq for Pol II (see Author response image 1) but we could not distinguish whether the remaining signal that we observed was due to the ~5% contaminating interphase cells in our mitotic prep. Furthermore, we also could not rule out potential effects of formaldehyde crosslinking on the mitotic ChIP. Instead, we turned to live-cell single molecule imaging. To address the reviewers’ comments, we redid all of our imaging experiments using a more sensitive dye and imaging set up (using far-red dye and lasers), and included Triptolide and Flavopiridol drug treatments with and without TBP degradation in interphase and mitotic cells. Also, instead of the table summarizing these results, we have included figures showing the fraction bound for each single cell that we analyzed (32 cells total, 8 technical replicates over 4 biological replicates – Figure 5A, B), and statistical analyses showing significant changes in the highlighted experiments. To summarize the results presented in the manuscript, we show that 1) our tagged Pol II is sensitive to inhibitors in both interphase and mitotic cells, suggesting that, even in mitosis, there exists a population of Pol II that is actively engaged; and 2) inhibitor treatment combined with TBP degradation during mitosis (Figure 5B, Mit+IAA vs. Mit+IAA+Flav, Mit+IAA+Trip) has no significant effect on the small remaining fraction of bound Pol II molecules, suggesting that the majority of *specific* Pol II interactions during mitosis is TBP-dependent. These results are discussed in the manuscript.

Also on the same topic, the data does not support the corresponding heading (Results, subsection “TBP recruits RNA Polymerase II to mitotic chromosomes”). This same team (Teves et al., 2016) showed that TBP fractionated with mitotic chromosomes, whereas Pol II did not (Figure 1—figure supplement 2 in Teves et al., 2016). In contrast, in the present study the authors show that in the heterozygous Halo-Pol II knock-in cell-line, the bound population of Halo-Pol II is 12.1% during mitosis (Results, subsection “TBP recruits RNA Polymerase II to mitotic chromosomes”, fourth paragraph). The authors should clarify this contradiction. Moreover, the effect of TBP degradation on Pol II recruitment and binding is quite mild, it changes from 12.1% in untreated mitotic cells to about 10.8% after TBP degradation. Is this change really significant? This very minor change does not support the claim that "TBP recruits RNA Polymerase II to mitotic chromosomes". The same comment stands for “Remarkably, such stable TBP binding leads to recruitment of Pol II to mitotic chromosomes, despite the general inhibition of transcriptional activity”. The authors should tone down their overstated conclusions.

Part 1 (“Also on the same topic, […] should clarify this contradiction.”). For several reasons, we disagree that our previous result (Teves et al., 2016) is necessarily a contradiction to our claims in the current manuscript. First, from the current study, we found that the small fraction of Pol II molecules that bind during mitosis has an average residence time of 5-8 seconds, making Pol II much more dynamic in mitosis compared to interphase (Figure 5B, C). This increase in dynamics likely affected Pol II during the mitotic chromosome purification in our 2016 study. The second reason relates to Part 2 of this comment. We agree that the majority of the ‘bound’ Pol II that we observe is largely non-specific interactions between Pol II and DNA/chromatin in general, and that the small fraction that changes when TBP is degraded is likely the fraction that specifically interacts with promoter DNA. Since promoters make up a very small fraction of the genome, it is not surprising that the number of Pol II molecules that specifically interacts with promoters in mitosis would be small. It is likely that what we would observe in the biochemical fractionation of mitotic chromosomes from our previous study is the stripping away of non-specific chromatin associations by our stringent washing and that our Western blots were not sensitive enough to detect the small fraction of specific binding events that the single molecule imaging was able to reveal.

Part 2 (“Moreover, the effect of TBP degradation […] tone down their overstated conclusions.”). After improving the imaging method (see response to Major concern point #3), our data now show that the average fraction of bound Pol II in mitosis is 15.39% (SEM 0.56) while after TBP degradation, the fraction bound decreases to 11.84% (SEM 0.51). After statistical analyses, this difference has a p-value < 0.05 (t-test). These results suggest that TBP does recruit a subset of Pol II molecules to mitotic chromosomes. However, we agree with the reviewers that we should be more careful in not overstating our conclusions, and we have made changes to the manuscript to reflect this. We have changed the heading of the section to “TBP recruits a subset of RNA Polymerase II molecules to mitotic chromosomes” and point out in the text that non-specific interactions dominate the fraction of bound Pol II in mitosis.

Finally, when analyzing the effects of mitotic TBP binding/depletion on the transcriptional reactivation rate (in Figure 6 or Figure 7), could the authors see any difference between Group 1, 2, 3 genes (as defined in Figure 3D)?

For the new analyses included in the revised manuscript, we performed differential peak analysis on the TBP ChIP-seq data using DiffBind (new Figure 3A, B) and found that the majority of the peaks in asynchronous and mitotic cells are not significantly different from each other (new Figure 3A and B), suggesting that the differences that we observed by k-means clustering (old Figure 3D), are minor level changes. Indeed, in our GO term analysis of the genes in these groups, we see similar classes of genes and we concluded that “the differences in TBP binding that we observe by ChIP-seq largely reflect the cyclical expression of genes during the cell cycle.” With this in mind, we performed the analysis requested by the reviewer, comparing the RNA data on Groups 1, 2, and 3 from the TBP ChIP-seq. We plotted the heatmap of the log_2_ ratio of reads in the 30- and 60-minute conditions relative to mitotically arrested (M30/M and M60/M, respectively), for both untreated and TBP-degraded samples using the 3 groups as defined in TBP ChIP-seq data. In general, we see no differences in the amount of RNA in groups 1 and 3, whereas group 2 shows much lower levels. Based on our GO term analysis of these groups, this seemed like the expected result. We have included this analysis as a new supplemental figure (Figure 7—figure supplement 2), and added the appropriate reference in the Results section of the main manuscript.

2) Need for further insight about which TBP-containing complex impacts on Pol II reactivation after mitosis. As TBP is involved in Pol I, Pol II, and Pol III transcription, the authors should show, which TBP-containing complex, SL1, TFIID and/or TFIIB, is playing the reactivation 'bookmarking role'. This would provide more novelty over published work.

This is an interesting question posed by the reviewers. We reasoned that our chrRNA-seq data may shed some light into this question. Upon further analysis of our chrRNA-seq data, we observe that tRNA genes show a large decrease in reactivation upon TBP depletion in mitosis, suggesting that RNA Polymerase III activity is also affected by TBP degradation in mitosis. We performed a similar analysis on ribosomal RNA genes (RNA Polymerase I-transcribed genes), but we see little change upon TBP depletion during mitosis. However, this result may reflect the stability of ribosomal RNA in mitotic cells. Because our assay relies on enrichment of nascent RNA through chromatin association, it is possible that effects of TBP degradation on nascent rRNAs may be masked in our assay. These results are now summarized in Figure 6—figure supplement 3 and in the Results section of the manuscript. However, we chose not to elaborate on these results as they are somewhat outside the scope of the current manuscript, but we will likely pursue these effects in future studies.

3) Asynchronous versus mitotic cells. Reviewers did not find sufficient details and controls about how the asynchronous versus mitotic cell populations were generated. Authors should demonstrate the degree of mitotic synchronization efficiency and the number of mitotic cells in the 'asynchronous' population. Otherwise the interpretation of all the ChIP-seq results becomes questionable. Alternatively, the authors should consider FACS sorting after synchronisation as described by Kadauke et al. (2012).

We apologize for this oversight. Our mitotic preparation methods are exactly the same as in the previously published 2016 study. To further quantify the purity of our mitotic preps using orthogonal methods, we imaged H2B-GFP after Nocodazole synchronization and shake off, and individually scored each cell as either mitotic or interphase (n=1075 cells). This data is now summarized in the new Figure 3—figure supplement 1, which shows that our mitotic prep typically achieves ~96% purity.